# LINGUINE: LearnIng to pruNe on subGraph convolUtIon NEtworks

## Abstract

Graph Convolutional Network (GCN) has become one of the most successful methods for graph representation learning. Training and evaluating GCNs on large graphs is challenging since full-batch GCNs have high overhead in memory and computation. In recent years, research communities have been developing stochastic sampling methods to handle large graphs when it is unreal to put the whole graph into a single batch. The performance of the model depends largely on the quality and size of subgraphs in the batch-training. Existing sampling approaches mostly focus on approximating the full-graph structure but care less about redundancy and randomness in sampling subgraphs. To address these issues and explore a better mechanism of producing high-quality subgraphs to train GCNs, we proposed the `Linguine` framework where we designed a meta-model to prune the subgraph smartly. To efficiently obtain the meta-model, we designed a joint training scenario with the idea of hardness based learning. The empirical study shows that our method could augment the accuracy of the current state-of-art and reduce the error incurred by the redundancies in the subgraph structure. We also explored the reasoning behind smart pruning via its visualization.

## 1 Introduction

Graph Representation Learning has attracted much attention from the research communities in recent years, with emerging new work every year. Graph Convolution Neural Networks (GCNs) were proposed as the extension of Convolutional Neural Networks(CNNs) (LeCun et al., 1995) on geometric data. The first spectral-based GCN was designed on Spectral Graph Theory (Bruna et al., 2013) and was extended by many following works (Henaff et al., 2015; Defferrard et al., 2016). Over recent years, the spatial-based counterpart (Kipf & Welling, 2016a) gained more attention and had facilitated many machine learning tasks (Wu et al., 2020; Cai et al., 2018) including semi-supervised node classification (Hamilton et al., 2017b), link prediction (Kipf & Welling, 2016b; Berg et al., 2017) and knowledge graphs (Schlichtkrull et al., 2018). In this work, we primarily focused on large-scale spatial-based GCNs (Hamilton et al., 2017a; Chen et al., 2018b; Gao et al., 2018; Huang et al., 2018; Zeng et al., 2019; Zou et al., 2019; Chiang et al., 2019), where a given node aggregates hidden states from its neighbors in the previous layer, followed by a non-linear activation to obtain the topological representation.

However, as the graph gets larger, GNN models suffer from the challenges imposed by limited physical memory and exponentially growing computation overhead. Recent work adopted sampling methods to handle the large volume of data and facilitate batch training. The majority of them could be classified as 3 types, layer-wise sampling (Hamilton et al., 2017a; Gao et al., 2018; Huang et al., 2018; Zou et al., 2019), node-wise sampling (Chen et al., 2018b) and subgraph sampling (Chiang et al., 2019; Zeng et al., 2019). In layer-wise sampling, we take samples from the neighbors of a given node in each layer. The number of nodes is growing exponentially as the GCNs gets deeper, which resulted in 'neighbor explosion'. In node-wise sampling, the nodes in each layer are sampled independently to form the structure of GCNs, which did avoid 'neighbor explosion'. But the GCN's structure is unstable and resulted in inferior convergence. In subgraph sampling, the GCNs are trained on a subgraph sampled on the original graph. The message was passed within the subgraph during training. This approach resolved the problem of neighbor explosion and can be applied to training deep GCNs. However, the subgraph's structure and connectivity had a great

impact in the training phase. It might result in suboptimal performance and slow convergence if the subgraph is overly sparse (Chen et al., 2018b). Different sampling methods can make a huge difference in the final accuracy and the convergence speed as shown in (Zeng et al., 2019). In the context of large-scale GCN training, the limitations in GPU memory makes the maximum batch size restricted. Research communities are actively seeking efficient sampling methods to deal with the challenges in scalability, accuracy, and computation complexity on large-scale GCNs.

The initiative of subgraph GCN is a stochastic approach to approximate their full-graph counterparts. However, experiments showed that GCN trained with partial information can achieve even less bias. (Zou et al., 2019) This was even more so when variance reduction is applied in GCN's training, which overcomes the negative effect induced by subgraph GCNs and improves its convergence speed as well as inference performance. (Hamilton et al., 2017a; Chen et al., 2018a; Zeng et al., 2019) However, the inner mechanism behind this random sample has yet to be studied.

GCNs are also made to be deeper and more complicated with architecture design. The model overcomes the drawbacks of the gradient vanishing problem via applying Deep CNN's residual/dense connection and dilated convolution, achieving state of art performance on open graph benchmarks. (Li et al., 2019; 2020; Weihua Hu, 2020) However, as their model is significantly larger (amounting to more than 100 layers in some occasions) than previous approaches, the model is suffered from high computation overhead and memory cost. With the model taking up much space, the batch sizes are also strictly limited. We aim to provide an easier solution to train complex models while maintaining a relatively large receptive field in the graph, and keeping the training quality.

We propose `Linguine` framework. In each forward pass, we 'smartly prune' inferior nodes to extract a concentrated smaller subgraph from the large subgraph randomly sampled previously. Therefore, we reduce the batch size and the memory requirement in training the model. We are also able to train the complex GCNs with larger receptive field and achieve better performance with the same budget. We parameterize the decision function in smart pruning with a light-weight meta-model, which is fed with the meta-information we obtained from training a light-weight proxy model. This keeps the extra cost of algorithm under control. Our framework is built upon existing subgraph sampling methods and utilize joint training to learn the meta-model. Our meta-model improved the quality of the subgraphs in training via actively dropping redundant nodes from its receptive field and concentrate the information.

We summarize the contributions of this work as follows:

1. We designed a new training framework `Linguine` which aims to train high-performance GCNs on large graphs via model-parameterized smart pruning techniques.
2. `Linguine` provides a joint-training algorithm called bootstrapping, originally designed to train the meta-model in smart pruning, but also has a favorable impact in augmenting existing models and converge to better solutions with scalability and lower bias. It is also an effective algorithm inspired by the idea in the real-world learning process.
3. Empirical study justified that `Linguine` framework worked well on different scaled public benchmarks and compares favorably to previous methods.
4. We did an analysis on the mechanism behind smart-pruning via graph visualization.

## 2 RELATED WORK

Our framework was inspired and built upon two popular branches in Machine Learning.

**Graph Neural Networks** Many GNNs have emerged over the recent years. Spatial-based GCNs are the most popular approaches among them and have gained broad interest from research communities (Atwood & Towsley, 2016; Niepert et al., 2016; Gilmer et al., 2017). They stack multiple graph convolutional layers to extract high-level representations. In each layer, every node in the graph aggregates the hidden states from its neighbors on the previous layer. The final output is the embedding of each node in the graph. Existing work on GCNs utilized sampling technique to perform efficient minibatch training. Common approaches can be categorized as *Layer-wise Sampling*, *Node-wise Sampling* and *Subgraph Sampling*. In *Layer-wise sampling* and *Node-wise Sampling*, the layers are sampled recursively from top to bottom to form mini-batches. The major difference between *Node-wise Sampling* and *Layer-wise Sampling* lied in the sampling mechanism.

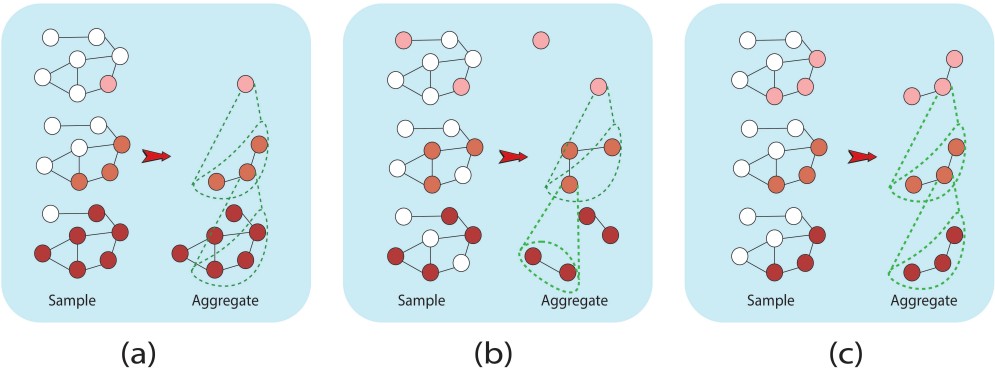

Figure 1: Illustration of Subgraph GCN: **(a)** Layer-wise Sampling **(b)** Node-wise Sampling **(c)** Subgraph Sampling

*Node-wise Sampling* sampled from the whole graph dataset whereas *Layer-wise Sampling* only focused on the neighbors from the upper layer.

In *Subgraph Sampling* we only sample once and propagate through the subgraph to formalize batch training. Among work in *Layer-wise Sampling*, GraphSAGE (Hamilton et al., 2017a) performed uniform Node-wise Sampling on the neighbors of nodes in the previous layer. (Ying et al., 2018) introduced the importance score to each neighbor and performed weighted aggregation to enhance the model. The potential drawback of *Layer-wise Sampling* is Neighbor Explosion where we created an overly-large receptive field in the bottom layer and resulted in memory overflow.

In *Node-wise Sampling*, FastGCN (Chen et al., 2018b) performs sampling independently across all layers and perform importance sampling on the node level to reduce the variance. This approach was further extended by LADIES (Zou et al., 2019) which performs importance sampling adaptively from top to bottom.

Recently, *Subgraph Sampling* emerged as a low complexity solution for large-scale GCN training. ClusterGCN (Chiang et al., 2019) performs clustering on the graph. This work partitions the graph into densely connected clusters. Random sampling is performed on the cluster level to form mini-batches in training. GraphSAINT (Zeng et al., 2019) directly sampled the mini-batches at node-level. This approach also performed normalization at the node level to reduce the sampling variance. It achieves the new state of the art performance on multiple public datasets. The main difference between *Subgraph Sampling* and other approaches is that GCN is trained on the subgraph instead of a massage passing flow containing different nodes on different layers.

Sampling subgraph has been an important method leveraging limited memory to achieve better performance. However, the variance introduced by sampling negatively impacts the training. Recent approaches adopted *Variance Reduction* (Hamilton et al., 2017a; Chen et al., 2018a; Zeng et al., 2019) to stochastically approximate the activations of sampled nodes. which was shown to boost the final model performance and convergence speed (Zeng et al., 2019). Besides, empirical study found that different sampling methods can also create a great gap in the final performance of GCNs and varied convergence speed.

The advance of deepening GCNs has drawn extensive attention in the research community, leveraging the computation power of high-performance GPU devices. (Li et al., 2019) adapted residual connections and dilated convolutions into GCNs. (Li et al., 2020) proposed a general aggregation function, which further improves the capacity of deep GCNs.

**Meta-Learning** Our work is inspired by recent meta-learning and model-teaching approaches, which is also a trendy topic in recent years.

Learning to Learn, or Meta-Learning (Schmidhuber, 1987; Thrun & Pratt, 2012) was proposed as an exploration of using automatic learning via transferring knowledge learned from meta tasks. It was designed as a two-level structure, in which the meta-level model evolves slowly and the task

model progresses quickly. Recently, meta-learning has been utilized in different machine learning scenarios. There is work trying to design optimizer and neural network architectures. (Andrychowicz et al., 2016; Li & Malik, 2016; Zoph & Le, 2016). Other approaches similar to Meta-Learning is Teaching (Anderson et al., 1985; Goldman & Kearns, 1992), which can be categorized as machine-teaching and hardness based methods. Machine teaching (Zhu, 2013) is to construct a minimal training set for the student model to learn a target model. However, due to the strong assumption of the oracle's existence in machine teaching. Hardness-based models are proposed on the assumption that a data order from easy to hard can benefit model training. There are many different works in the hardness-based method. Curriculum Learning (CL) (Bengio et al., 2009; Spitkovsky et al., 2010; Graves et al., 2017) utilizes a hardness measure based on a heuristic understanding of data. Self-paced learning(SPL) (Kumar et al., 2010) heuristically define the scheduling of training data in an easy-to-hard manner. Learning to Teach (Fan et al., 2018; Wu et al., 2018) designed a new teacher-student model and perform transferable model teaching.

Our meta-model incorporates graph topological information and node-wise performance information into meta information. We used a learned model instead of heuristic and fixed rules in the model to explore the optimal policy.

## 3 METHODS

In this section, we introduce the background of our problem, followed by the formal definition of $Linguine$ framework.

### 3.1 PRELIMINARIES

A GCN learns representation of an un-directed, arributed graph $\mathcal{G}(\mathcal{V}, \mathcal{E})$ where each node $\boldsymbol{v} \in \mathcal{V}$ has attribute $\boldsymbol{x}_v \in \mathbb{R}^f$. Let $\boldsymbol{A}$ be the adjacency matrix and $\tilde{\boldsymbol{A}}$ be the normalized one. (i.e., $\tilde{\boldsymbol{A}} = \boldsymbol{D}^{-\frac{1}{2}} \boldsymbol{A} \boldsymbol{D}^{-\frac{1}{2}}$, where $\boldsymbol{D}$ is the diagonal degree matrix). The activation of node $v$ is $\boldsymbol{x}_v^{(l)} \in \mathbb{R}^{f^{(l)}}$, and the weight matrix is $\boldsymbol{W}^{(l)}$. Note that $\boldsymbol{x}_v = \boldsymbol{x}_v^{(1)}$. The rules of forward propogation per layer is defined as:

$$\boldsymbol{x}_v^{(l+1)} = \sigma\Big( \sum_{u \in \mathcal{V}} \tilde{\boldsymbol{A}}_{v,u} W^{(l)} \boldsymbol{x}_u^{(l)} \Big) \tag{1}$$

where $\tilde{\boldsymbol{A}}_{v,u}$ is the scalar element of $\tilde{\boldsymbol{A}}$. And $\sigma$ is the activation function.

The propogation of Subgraph GCN can be summarized as:

$$\hat{\boldsymbol{x}}_v^{(l+1)} = \sigma\Big( \sum_{u \in \mathcal{V}} \hat{\boldsymbol{A}}_{v,u} W^{(l)} \hat{\boldsymbol{x}}_u^{(l)} \Big) \tag{2}$$

$\hat{\boldsymbol{A}}_{u,v}$ is the stochastic adjacency martix we sampled previously. It is also a mapping in $\mathbb{U} \to \mathbb{V}$ where $\mathbb{U}$ and $\mathbb{V}$ are subspaces of hidden-state space of layer $l$ and $l+1$ respectively. The $\hat{\boldsymbol{x}}_u$ is the $l$-layer embeddings of sampled nodes. As have been addressed previously on the 3 different types of subgraph GCNs, the main diference can be formally defined with $\hat{\boldsymbol{A}}_{u,v}$: In **node-wise sampling**, $\hat{\boldsymbol{A}}_{u,v}$ is stochastically sampled from $\boldsymbol{A}$ with $|\mathbb{U}|$ rows and $|\mathbb{V}|$ columns, where the number of nodes is not necessarily identical. In **layer-wise sampling**, the $\hat{\boldsymbol{A}}_{u,v}$ is similar to that in node-wise sampling where the cross-section of those rows and columns is 1. Meanwhile, the set $\mathbb{U}$ is a subset of $\mathbb{V}$, making $\hat{\boldsymbol{A}}_{u,v}$ a surjective mapping. In **subgraph sampling**, the set $\mathbb{U}$ and $\mathbb{V}$ are identical, which implied that the same adjacency matrix $\hat{\boldsymbol{A}}_{u,v}$ in every layer. In this case, the mapping is isotonic, which keeps the network structure stable and facilitates the deepening of GCNs.

The goal of training a Subgraph GCN is to perform batch training with each batch containing a subset of $\mathcal{V}$ and approximate the effect of training a full graph GCN.

The learning tasks can be categorized as inductive and transductive. In the inductive setting, we assume neither attributes nor connections of the test node can be seen in the training phase whereas in the transductive setting we have access to them. The inductive model is much more challenging to train as shown in (Hamilton et al., 2017a) and has to generalize to unseen graphs.

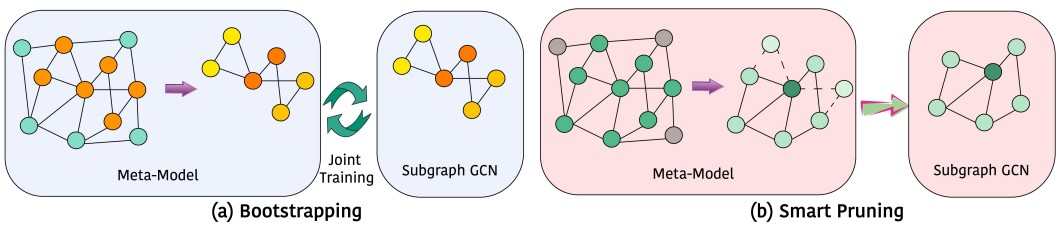

Figure 2: Linguine Framework

## 3.2 LINGUINE FRAMEWORK

Linguine follows the design philosophy of utilizing a Meta-Model to optimize the training of subgraph GCNs. We divide the whole framework into 2 consecutive parts: Bootstrapping and Smart Pruning. In **Bootstrapping** we designed a joint training scenario detailed in 3.2.1 and algorithm 1 .We also analyzed the convergence of it under the convex scenario. In **Smart Pruning** we designed algorithm 2 and detailed the mechanism in 3.2.2

Our goals are set as follows: **(1)** Optimizing the subgraphs to minimize information loss and achieve better generalization. **(2)** Leveraging limited memory to achieve scalable learning **(3)** Utilizing the model feedback to optimize the training process.

---

**Algorithm 1** Bootstrapping

**Input** Full Graph $\mathcal{G}(\mathcal{V}, \mathcal{E}, \boldsymbol{X})$, light-weight GCN $G_{light}$
**Output** Learned Meta-Model $M$
1: Initialize meta-information $\boldsymbol{I}(\mathcal{V})$ , meta-model weight $W_M$, $G_{light}$
2: **for** minibatch $i$ from 0 to $N$ **do**
3:   Randomly Sample a subgraph $\mathcal{G}_{sample}(\mathcal{V}_{sample}, \mathcal{E}_{sample}, \boldsymbol{X}_{sample}) = sample(\mathcal{G})$
4:   Fed $M$ with $\boldsymbol{I}(\mathcal{V}_{sample})$ and get pruning weight $\mathcal{W}$
5:   Normalize $\mathcal{W}$ and get $\hat{\mathcal{W}}$
6:   $\boldsymbol{X}_{sample} = \boldsymbol{X}_{sample} \odot \hat{\mathcal{W}}$
7:   Fed $G_{light}$ with $\mathcal{G}_{sample}$ and calculate task loss $L$
8:   **if** $i$ is even **then**
9:     Calculating $\frac{\partial L}{\partial W_{G_{light}}}$ and update $G_{light}$
10:   **else**
11:     Calculating $\frac{\partial L}{\partial W_M}$ and update $M$
12:   **end if**
13:   Update $\boldsymbol{I}(\mathcal{V}_{sample})$
14: **end for**

---

### 3.2.1 BOOTSTRAPPING

In algorithm 1, the meta-model is designed as a 2-layer vanilla GCN. The light-weight GCN (i.e. $G_{light}$) is used as a proxy, escalating the training of meta-model. The meta-information (i.e. $\boldsymbol{I}(\mathcal{V})$) includes 3 types of information from the model feedback: (Fan et al., 2018): **(1)** Model information (i.e. average loss in the past epochs in training set, the loss oberserved so far on validation set ) **(2)** Node information (i.e. nodes' predicted labels) **(3)** Joint information (i.e. predicted class probabilities of Subgraph GCN in previous epochs)

The meta-model is jointly trained with a lightweight GCN where the update of the meta-model and the light-weight GCN are trained alternately. The goal of bootstrapping is to obtain a potent meta-model selective of existing subgraphs with the most informative nodes while weakening the redundant nodes. The light-weight design expedites the training process since it is budget-friendly. The sampling algorithm is user-defined. We use random-node subgraph sampling in our experiments, which is the simplest and most cost-effective.

---

**Algorithm 2** Smart Pruning

---

**Input** Full Graph $\mathcal{G}(\mathcal{V}, \mathcal{E}, \boldsymbol{X})$, Meta-model $M$, Meta-information $\boldsymbol{I}(\mathcal{V})$, Desired Subgraph Scale $K$
**Output** High Performance GCN $G$

 1: Initialize GCN weight $W_G$
 2: **for** each minibatch **do**
 3:    Randomly sample a subgraph $\mathcal{G}_{sample}(\mathcal{V}_{sample}, \mathcal{E}_{sample}, \boldsymbol{X}_{sample})$ from $\mathcal{G}$
 4:    Feed $M$ with $\boldsymbol{I}(\mathcal{V}_{sample})$ and obtain pruning score $\mathcal{W}$
 5:    Select top-$K$ nodes in $\mathcal{V}$ and prune the rest, forming $\mathcal{G}_{prune}$
 6:    Normalize top $K$ score and drop the rest, getting $\hat{\mathcal{W}}_{topK}$
 7:    $\boldsymbol{X}_{prune} = \boldsymbol{X}_{prune} \odot \hat{\mathcal{W}}_{topK}$
 8:    Feed $G$ with $\mathcal{G}_{prune}$ and calculate task loss $L$
 9:    Backpropogate with $\frac{\partial L}{\partial W_G}$ and update $G$
10:    Update $\boldsymbol{I}(\mathcal{V}_{prune})$
11: **end for**

---

Formally, the update of model weight of minibatch $i$ can be explicitly written as (For simplicity, we take fixed step vanilla gradient descent as scenario):

$$w = w - \eta \nabla_M L_i(W) \tag{3}$$
$$w = w - \eta \nabla_G L_i(W) \tag{4}$$

$\nabla_G$ and $\nabla_M$ are the partial gradients on GCN and meta-model respectively. Since the parameters of GCN and meta-model are independent of each other, formally, $\langle \nabla_M, \nabla_G \rangle = 0$.

We then provide a theorem on the convergence on this joint training scenario in convex scenario and leave the proof in the appendix.

**Theorem 1 (Convergence of Joint Training)** *Suppose the function $f : \mathbb{R}^n \to \mathbb{R}$ is convex and differentiable, and that its gradient is Lipschitz continuous with constant $L > 0$. Then if we run gradient descent for k iterations with a fixed step size $\eta \leq \frac{1}{L}$ under the joint training theme, it will yield a solution $f(x^{(t)})$ which satisfies:*

$$f(x^{(t)}) - f(x^*) \leq \frac{\|x^{(0)} - x^*\|_2^2}{\eta t} \tag{5}$$

*where $f(x^*)$ is the optimal value. This means that joint training theme of convex function is guaranteed to converge and it converges with rate $O(1/t)$.*

### 3.2.2  SMART PRUNING

In Smart Pruning, the meta-model is fixed and acts as a scorer on the previously sampled subgraph. We keep the nodes with top $K$ scores in the subgraph while pruning the rest before forwarding it to the target GCN. The meta-model is then acting like a teacher akin to that of  (Wu et al., 2018; Kumar et al., 2010), who selects the proper 'material' (e.g. subgraphs) for the student model (e.g. complex GCNs) to learn. The meta-model is still fed with the meta-information from the student model alike Bootstrapping. This is akin to teachers getting feedback (i.e. what has been well received and what has not) from their students and adaptively changing the teaching contents for the next lecture in the real world. Since the full graph is overly large to digest by the student model, the meta-model is also acting as a concentrator and enlarges the receptive field while keeping a relatively small mini-batch size. This is done through dropping the redundant nodes from the existing subgraphs by pruning, our assumptions of redundancy are characterized as: **(1)** Nodes that are hard to learn by existing model (i.e. having high training loss) **(2)** Nodes that affect its neighbors' categorization. (i.e. nodes with high degree) These two assumptions have been discussed in the experiment visualization at Appendix D.

## 4  EXPERIMENTS

The purpose of our experiments is to answer the following questions: (1) How would bootstrapping algorithm compare with original methods? (2) How would graph sampling methods affect the

| Datasets | PPI | Flickr | Reddit | Yelp | OGB-Product |
|---|---|---|---|---|---|
| Vanilla GCN | 0.515(0.006) | 0.492(0.003) | 0.933(0.000) | 0.378(0.001) | 0.756(0.002) |
| GraphSAGE | 0.637(0.006) | 0.501(0.013) | 0.953(0.001) | 0.634(0.006) | 0.785(0.001) |
| FastGCN | 0.513(0.032) | 0.504(0.001) | 0.924(0.001) | 0.265(0.053) | - |
| ClusterGCN | 0.875(0.004) | 0.481(0.005) | 0.954(0.001) | 0.609(0.005) | 0.790(0.003) |
| GraphSAINT-RN | 0.960(0.001) | 0.507(0.001) | 0.962(0.001) | 0.641(0.000) | 0.781(0.002) |
| GraphSAINT-RW | 0.981(0.004) | 0.511(0.001) | 0.966(0.001) | 0.653(0.003) | 0.791(0.002) |
| DeepGCN | - | - | - | - | **0.810**(0.002) |
| RW + Bootstrapping | **0.988**(0.003) | **0.517**(0.002) | **0.974**(0.001) | **0.680**(0.004) | 0.803(0.002) |
| RN + Bootstrapping | 0.980(0.004) | 0.512(0.003) | 0.970(0.003) | 0.672(0.003) | 0.790(0.002) |

Table 1: F1-micro benchmark on node classification task

outcome of bootstrapping? (3) How many nodes should we prune to achieve commendable performance? (4) How does the pruning work?

We address the above problem by comparing the different algorithmic compositions of bootstrapping with the current state of arts as well as the vertical comparison within the pruning algorithm. The code we implemented in Pytorch is available online[1].

### 4.1 DATASETS

**PPI**: Protein-protein interaction networks from (Zitnik & Leskovec, 2017). Our goal is classifying protein functions based on the interactions between human tissue proteins. **Flickr**: Flickr dataset from (Zeng et al., 2019) We did categorization on the images based on the properties of online images.**Reddit**: Reddit dataset comes from (Hamilton et al., 2017a), containing Reddit posts collecting from different communities. We did a prediction on the communities of online posts using user comments. **Yelp**: Yelp dataset comes from (Zeng et al., 2019), containing relationships between custom reviewers, on which we did categorization of types of businesses with customer reviewers and their friendship. **Open Graph Benchmarks**: (OGB for short) is a diverse set of challenging benchmark datasets (Weihua Hu, 2020). We used **Ogbn-proteins** where the major task is to categorize the protein based on the associations between different proteins. **Ogbn-products** comes from the Amazon product co-purchasing network. This dataset is also used in (Chiang et al., 2019) but it is modified on the split ratio to produce a much more challenging task. We did a classification task on the products' category. Dataset statistics are in Table 2

### 4.2 BASELINES

We used normally trained large-scale GCN as baselines for comparison: **Vanilla GCN**[2] (Kipf & Welling, 2016a) is the simplest spatial-based GCN. **GraphSage**[3] (Hamilton et al., 2017a) is a typical layer-wise sampling method. **FastGCN**[4] (Chen et al., 2018b) is a typical node sampling method. **ClusterGCN**[5] The first subgraph-sampling GCN algorithm. **GraphSAINT** (Zeng et al., 2019) is another typical subgraph sampling method and the state of art method. **DeepGCNs** (Li et al., 2019; 2020) deepens the GCNs and is by far the most parameterized GCN, which scales up to 8 to 10 times the parameter of previous GCNs in some cases with 10 times the layer of regular one. It is also straining the memory budget of GPUs and requires very high computation power with long convergence time.

### 4.3 RESULTS

**Bootstrapping** We examine the effect of doing joint training in the bootstrapping phase. Our implementation of GCN is a 2 layer SageNet. Hamilton et al. (2017a) Random node(**RN** for short)

---

[1]`https://github.com/anonym-code/LINGUINE`

[2]`https://github.com/tkipf/gcn`

[3]`https://github.com/williamleif/GraphSAGE`

[4]`https://github.com/matenure/FastGCN`

[5]`https://github.com/benedekrozemberczki/ClusterGCN`

and random walk(**RW** for short) as sampling algorithm respectively. Our experiments show that the Bootstrapping algorithm can achieve state-of-art performance on public datasets. All baselines are reported at their best parameter settings on 2-layer GCN as has been studied extensively in (Zeng et al., 2019). We also compared the influence of different batch sizes in Bootstrapping in the appendix C. The result shows that our joint training strategy has a bigger improvement over existing methods on larger datasets. However, as has been discussed in 3.2.1, the joint training theme has lower convergence speed in experiments, roughly using 1.5 times the number of epochs to converge than the one trained directly, this can be partially justified in the theoretical results we give in the theorem 1.

**Smart Pruning** We did experiments on OGB-Protein on a 24 layer DeepGCN backbone. The result is Figure 3. The implementation is using a 2-layer vanilla GCN as the light-weight proxy model in the algorithm 2. We adjust different batch sizes as well as pruning ratio and extensively studied the effect of smart pruning. We found that smart pruning could improve the generalization of complex GCN model when the batch sizes are limited and achieve commendable performance. One interesting symptom is that we achieve the best model performance when the pruning ratio around 0.8. We can achieve even performance with the original at 0.5. This implies that we do not need the whole subgraph as the training set to achieve the optimal generalization. The pruning process removed some of the redundancies and helped the model to 'focus' on the key elements in the graph. However, when we continued to prune more nodes from the subgraph, the performance dropped. As the pruning ratio goes lower, more nodes are out of reach, which sparsifies the graph and caused bad generalization. We did an extensive visualization of the pruning process to explore the mechanism behind it. It shows that the first batch of nodes we pruned out are those with high loss and high degrees. These nodes are difficult for the model to learn since the background knowledge is over-complicated as their adjacent nodes might belong to totally different classes. Learning these nodes is also complicating the learning process. This explanation follows the idea that in the real world, the learning process is a hardness-based process for human beings. We provide a more detailed analysis in the appendix D.

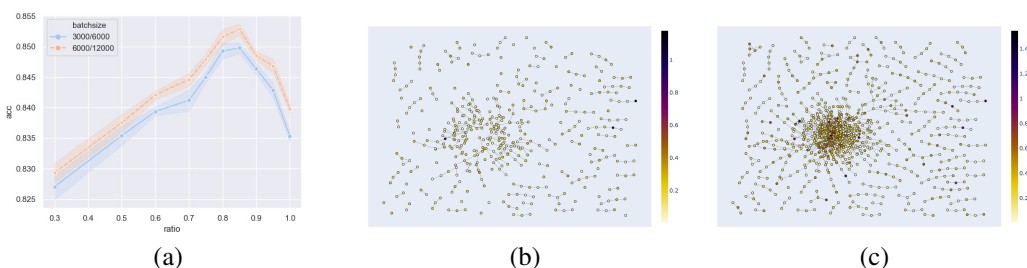

(a)                              (b)                              (c)

Figure 3: **(a)** is the test result of **OGB-Protein** on smart pruning with different batch sizes and pruning ratio, we used two settings of bootstrapping/smart pruning batch sizes configuration. Means and standard deviation is reported accordingly. Experiments are based on 20 runs of the same algorithm with different initialization. **(b)** One visualization of the real-time pruned subgraph during training at 0.7 pruning ratio on Yelp dataset. Color bar is of loss scale. The original subgraph size is 1000.**(c)** The original subgraph is in (b)

## 5    CONCLUSION AND FUTURE WORK

We have presented `Linguine` as a subgraph GCN training framework. Our design on bootstrapping and smart pruning algorithm improved the quality of GCN models. Extensive empirical study and visualization of the pruning procedure had justified the rationale of pruning subgraphs. Future directions included incorporating such mechanisms into designing better sampling methods in training and the study of training GCNs with partial information from the original graphs to achieve better generalization and efficiency.

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

## A  PROOF OF CONVERGENCE

We give a detailed proof of convergence on the joint training theme. We had the assumption that $\nabla f$ is $L$-Smooth and $\langle \nabla_D, \nabla_M \rangle = 0$ which implies that $\nabla_M f$ and $\nabla_D f$ is Lipschitz continuous with constant $L$ and $\eta_t \equiv \eta < \frac{1}{L}$ and

$$\|\nabla_M f\|_2^2 + \|\nabla_D f\|_2^2 = \|\nabla f\|_2^2 \tag{0}$$

In the joint training we can derive the bound of $\Delta_{t+1} = f(x^{t+1}) - f(x^t)$ as

$$f(x^{t+1}) - f(x^t) \leq \nabla_i f(x^t)^\top (x^{t+1} - x^t) + \frac{L}{2}\|x^{t+1} - x^t\|_2^2$$

$$= -\eta_t \|\nabla_i f(x^t)\|_2^2 + \frac{\eta_t^2 L}{2}\|\nabla_i f(x^t)\|_2^2$$

$$\leq -\frac{\eta}{2}\|\nabla_i f(x^t)\|_2^2 \tag{1}$$

where $i = M$ when $t$ is odd and $i = D$ when $t$ is even.

It follows that

$$\|x^{t+1} - x^*\|_2^2 = \|x^t - x^* - \eta(\nabla_i f(x^t) - \nabla_i f(x^*))\|_2^2$$

$$= \|x^t - x^*\|_2^2 - 2\eta\langle x^t - x^*, \nabla_i f(x^t) - \nabla_i f(x^*)\rangle + \eta^2\|\nabla_i f(x^t) - \nabla_i f(x^*)\|_2^2$$

$$\leq \|x^t - x^*\|_2^2 - \frac{2\eta}{L}\|\nabla f(x^t) - \nabla f(x^*)\|_x^2 + \eta^2\|\nabla_i f(x^t) - \nabla_i f(x^*)\|_2^2$$

$$\leq \|x^t - x^*\|_2^2 - \eta^2\|\nabla_i f(x^t) - \nabla_i f(x^*)\|_2^2$$

$$= \|x^t - x^*\|_2^2 - \eta^2\|\nabla_i f(x^t)\|_2^2$$

where $i = M$ when $t$ is odd and $i = D$ when $t$ is even. This is followed by

$$\|x^{t+2} - x^*\|_2^2 \leq \|x^t - x^*\|_2^2 - \eta^2\|\nabla f(x^t)\|_2^2 \tag{2}$$

according to (1).

Using Convexity and Cauchy-Schwarz inequality we obtain:

$$f(x^*) - f(x^t) \leq \nabla_i f(x^t)^\top (x^* - x^t) \leq -\|\nabla_i f(x^t)\|_2 \|x^t - x^*\|_2 \tag{3}$$

It follows (2) by

$$\|\nabla f(x^t)\|_2^2 = \|\nabla_M f(x^t)\|_2^2 + \|\nabla_D f(x^t)\|_2^2 \geq 2\Big(\frac{f(x^t) - f(x^*)}{\|x^t - x^*\|_2}\Big)^2 \geq 2\Big(\frac{f(x^t) - f(x^*)}{\|x^0 - x^*\|_2}\Big)^2 \tag{4}$$

Setting $\Delta_t := f(x^t) - f(x^*)$ and combing the above bounds (1)(2)(3)(4) yield

$$\Delta_t - \Delta_t \leq -\frac{\eta}{\|x^0 - x^*\|_2^2}\Delta_t^2$$

Dividing both sides by $\Delta_t \Delta_{t+1}$ and rearranging terms give

$$\frac{1}{\Delta_{t+1}} \geq \frac{1}{\Delta_t} + \frac{\eta}{\|x^0 - x^*\|_2}\frac{\Delta_t}{\Delta_{t+1}}$$

$$\geq \frac{1}{\Delta_t} + \frac{\eta}{\|x^0 - x^*\|_2}$$

$$\geq \frac{1}{\Delta_1} + \frac{\eta t}{\|x^0 - x^*\|_2}$$

Using the fact that $f$ is decreasing on every iteration, we have

$$\frac{1}{\Delta_t} \geq \frac{\eta t}{\|x^0 - x^*\|_2}$$

and

$$f(x^t) - f(x^*) \leq \frac{\|x^0 - x^*\|_2^2}{\eta t}$$

as claimed.

## B    DATASET STATISTICS

| Datasets | Nodes | Edges | Feature | Classes | Class Type | Train/Val/Test |
|----------|-------|-------|---------|---------|------------|----------------|
| PPI | 14,755 | 225,270 | 50 | 121 | Multi | 0.66/0.12/0.22 |
| Flickr | 89,250 | 889,756 | 500 | 7 | Single | 0.50/0.25/0.25 |
| Reddit | 232,965 | 11,606,919 | 602 | 41 | Single | 0.66/0.10/0.24 |
| Yelp | 716,847 | 6,977,410 | 300 | 100 | Multi | 0.75/0.10/0.15 |
| ogbn-protein | 132,534 | 39,561,252 | 8 | 112 | Multi | 0.65/0.16/0.19 |
| ogbn-product | 2,449,029 | 61,859,140 | 100 | 47 | Single | 0.081/0.016/0.903 |

Table 2: Statistics for datasets

## C    HYPERPARAMETER EVALUATION

Since we found that batch size of subgraphs greatly determines the actual impacts the performance of GCNs, we explored the results trained with different batchsizes. This showcased it is important that we made the right choice on the batch sizes in training large graph GCNs.

| Sampler | Batch Size | PPI | Yelp | Flickr | Reddit |
|---------|------------|-----|------|--------|--------|
| Random Node | 2000 | 0.901 | 0.573 | 0.470 | 0.933 |
| | 4000 | 0.942 | 0.587 | 0.485 | 0.954 |
| | 6000 | 0.958 | 0.595 | 0.504 | 0.970 |
| | 8000 | 0.972 | 0.603 | 0.508 | 0.969 |
| | 10000 | 0.980 | 0.648 | 0.512 | - |
| | 12000 | 0.976 | 0.672 | 0.511 | - |
| Random Walk | 2000 | 0.934 | 0.584 | 0.496 | 0.962 |
| | 4000 | 0.983 | 0.623 | 0.505 | 0.969 |
| | 6000 | 0.962 | 0.642 | 0.506 | 0.974 |
| | 8000 | 0.982 | 0.663 | 0.513 | 0.972 |
| | 10000 | 0.988 | 0.647 | 0.517 | - |
| | 12000 | 0.987 | 0.682 | 0.514 | - |

Table 3: Detailed Result of Bootstrapping on various batch sizes

## D    VISUALIZATION OF SUBGRAPHS IN TRAINING

We take the snapshots of subgraphs before and after smart pruning and visualized them together with the degree histogram as well as loss histograms in 9. We used batch size 1000 for clarity of figures and showcased the smart-pruned subgraphs at different levels of pruning ratio and training epochs. The figure shows that our pruning method is especially good at selecting low-loss nodes during training and simplifying the graph structure. Also, as the training proceeds, the pruned graph became more sparse with most low-loss nodes exists. The performance of GCNs are the best in the ratio of $0.9$ and $0.7$ can also be attested in the visualization since we preserved most of the details in the graph and cut off nodes concentrating at a 'high loss hull'. However, when the pruning ration

continues to go lower, where we preserve less structure within the graph, the performance didn't seem to improve anymore but drops monotonically. This can be caused by less information in the graph, as many nodes are ignored in the training process. It is also surprising that we can maintain more than 90% accuracy with only 30% nodes in the graph.

We can conclude that, since the model has a strong tendency to overfit at the verge of different classes, removing some training data through meta-learning could be an effective approach especially in geometric deep learning, where the graph structure itself determines the network structure. We have seen work adding up more and more structure (more complicated aggregation functions, more layers .etc) to achieve better model capacity on learning large graphs, however, this work addresses the problem from a different direction, rather than adding up, we aim to remove the redundancies. On the other hand, we also take the message that GCN itself might also be an over-parameterized on large graphs.

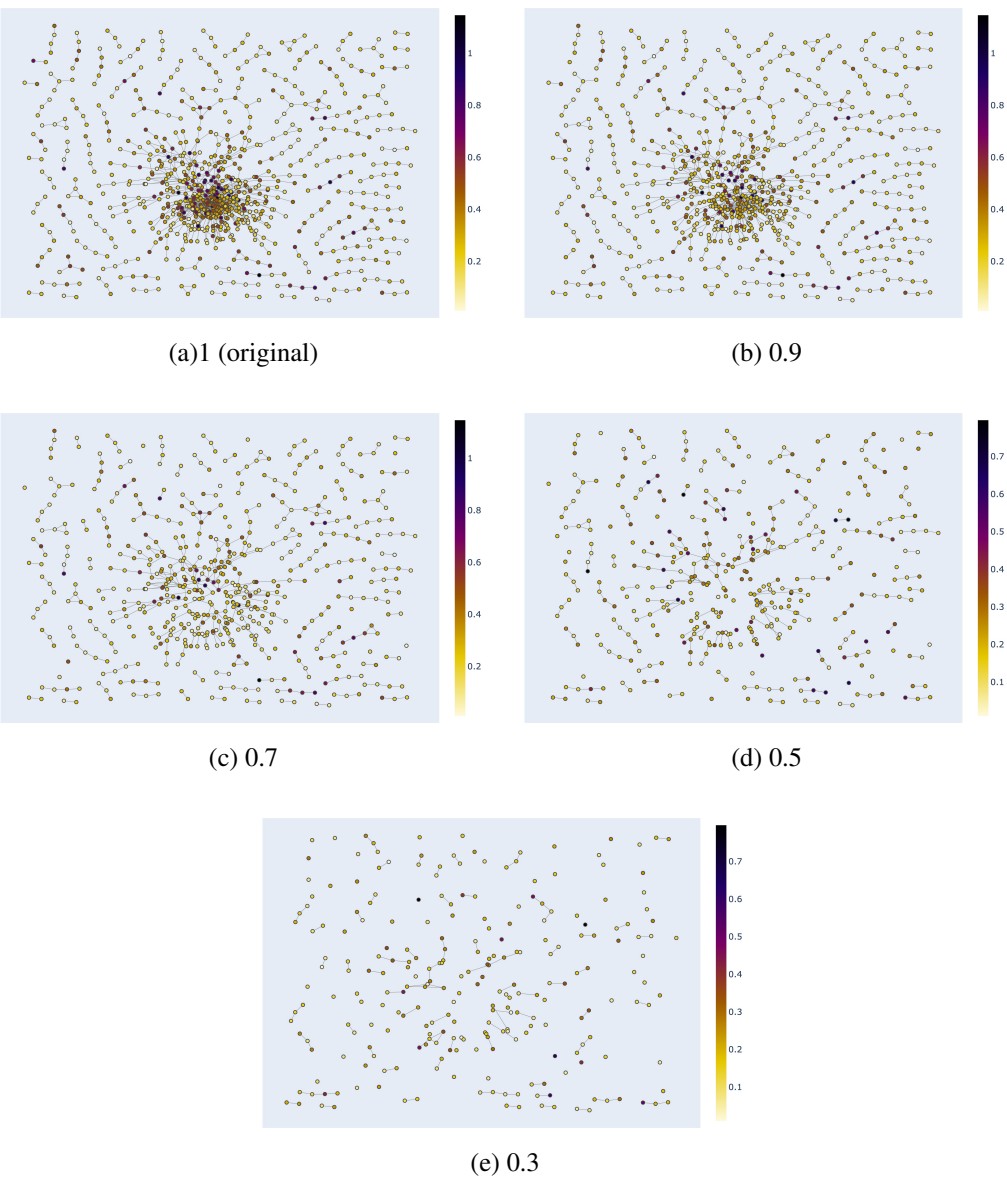

Figure 4: Pruning example on 1000 node subgraph of Yelp training with different ratio at epoch 100, color on loss scale

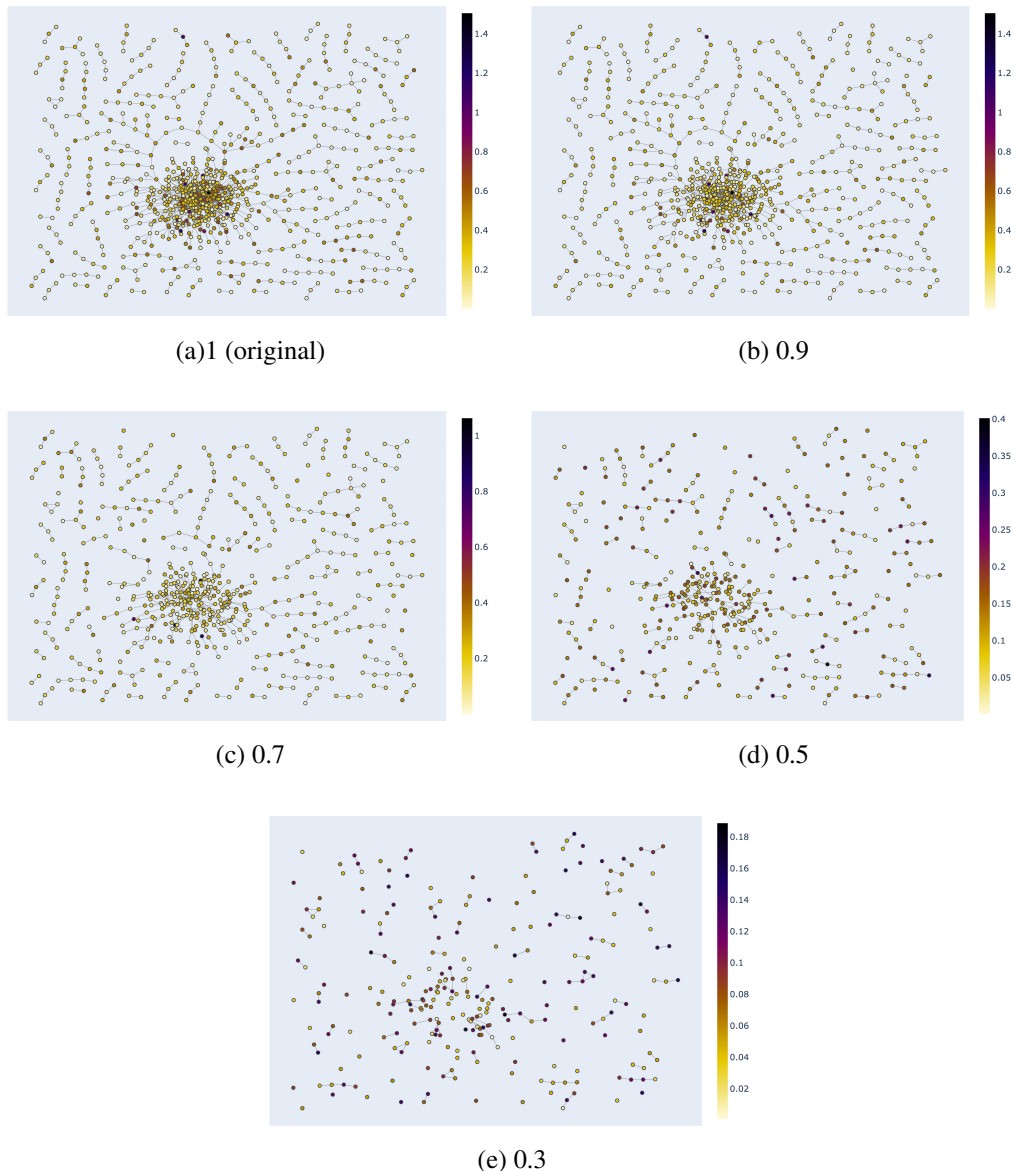

Figure 5: Pruning example on 1000 node subgraph of Yelp training with different ratio at epoch 500, color on loss scale

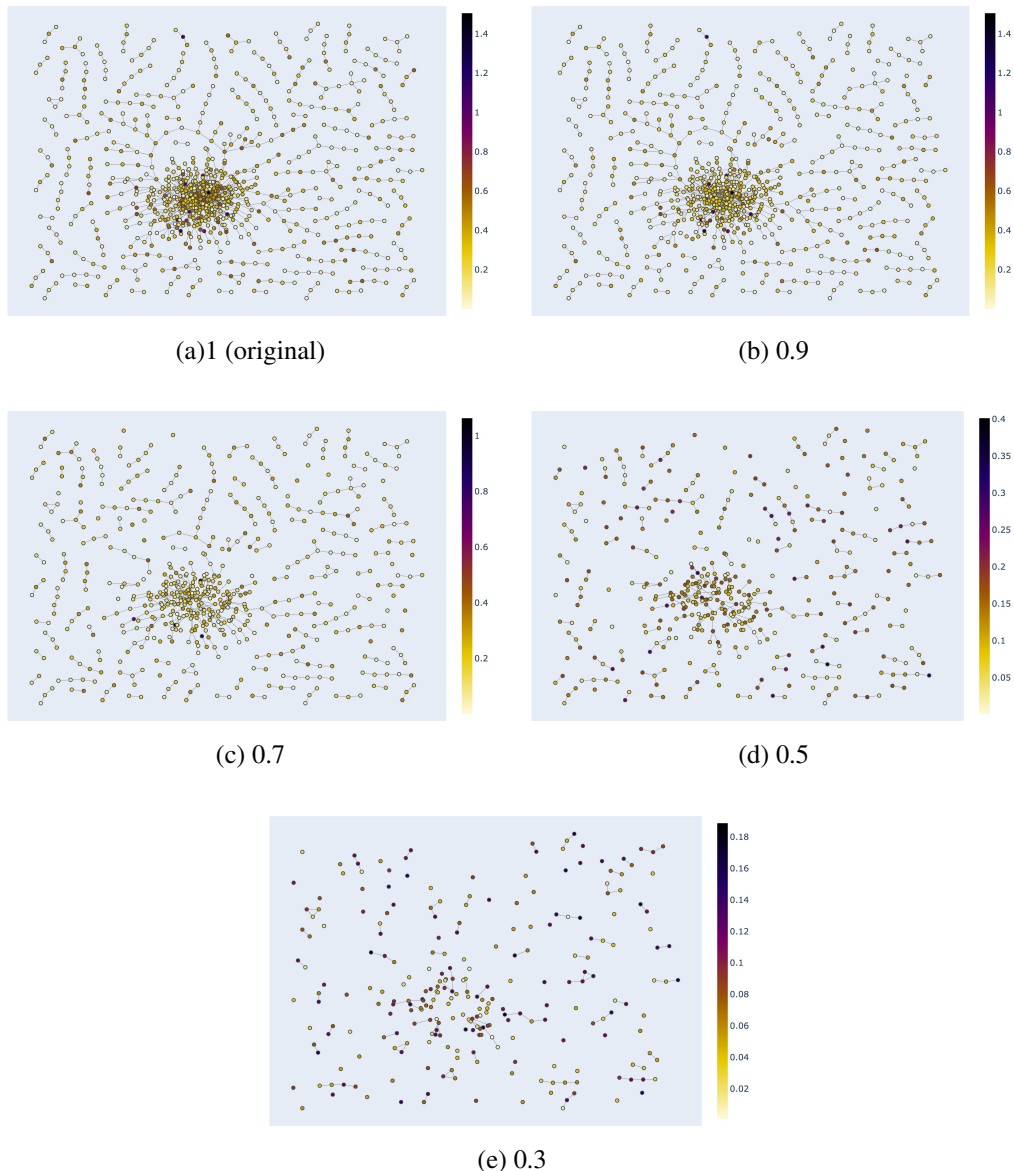

(a)1 (original)

(b) 0.9

(c) 0.7

(d) 0.5

(e) 0.3

Figure 6: Pruning example on 1000 node subgraph of Yelp training with different ratio at epoch 500, color on loss scale

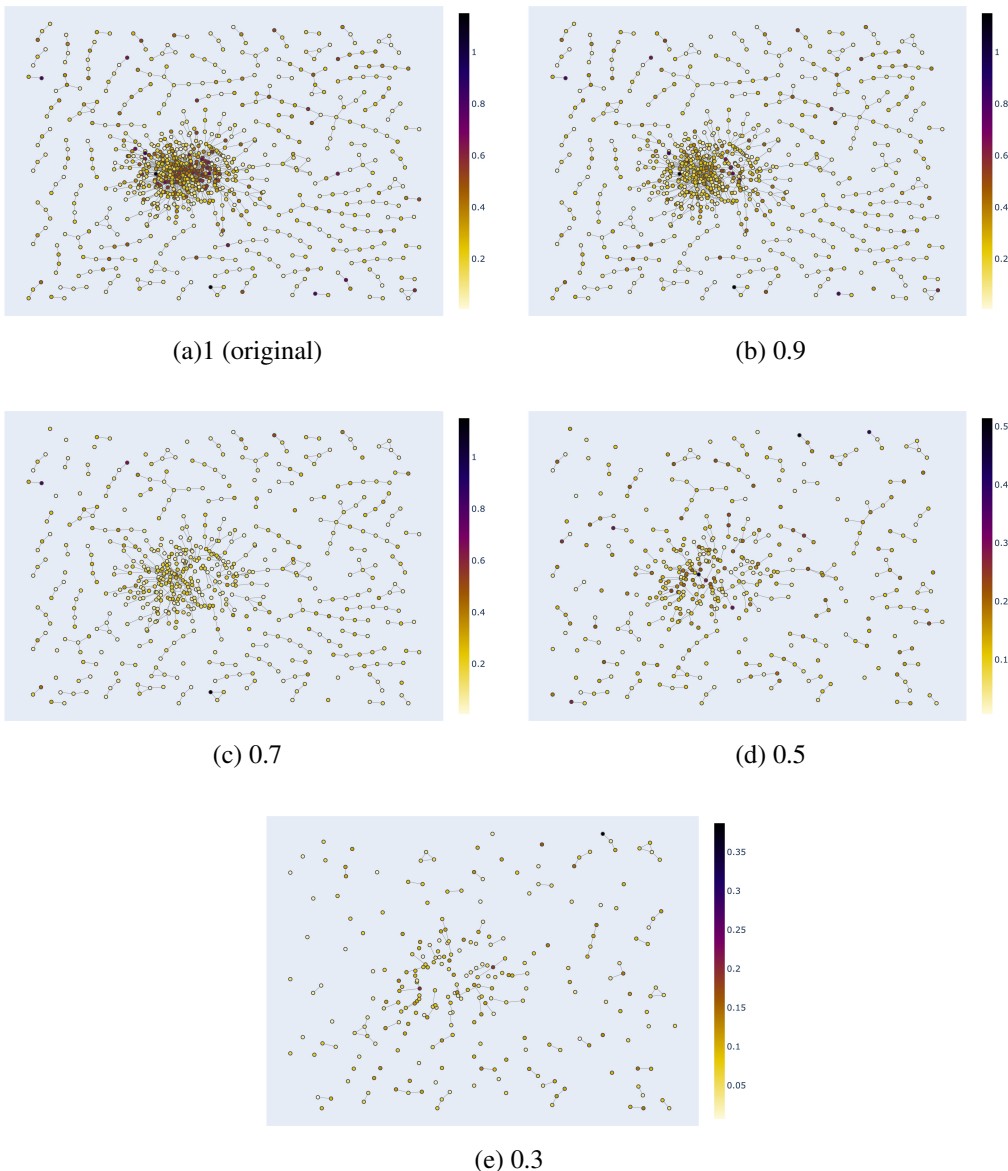

Figure 7: Pruning example on 1000 node subgraph of Yelp training with different ratio at epoch 700, color on loss scale

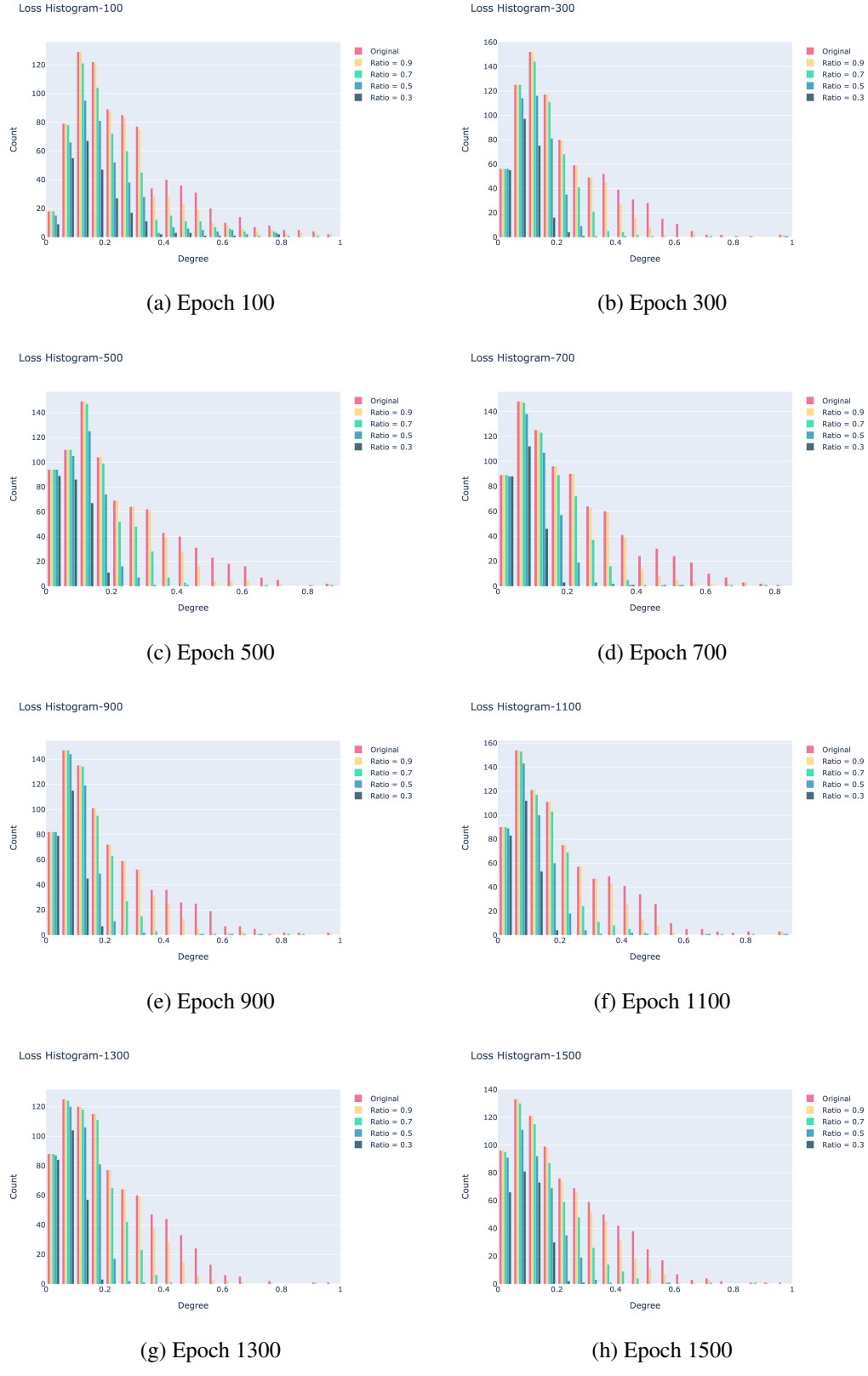

Figure 8: Loss Histogram of Flickr dataset at different epochs and pruning ratio

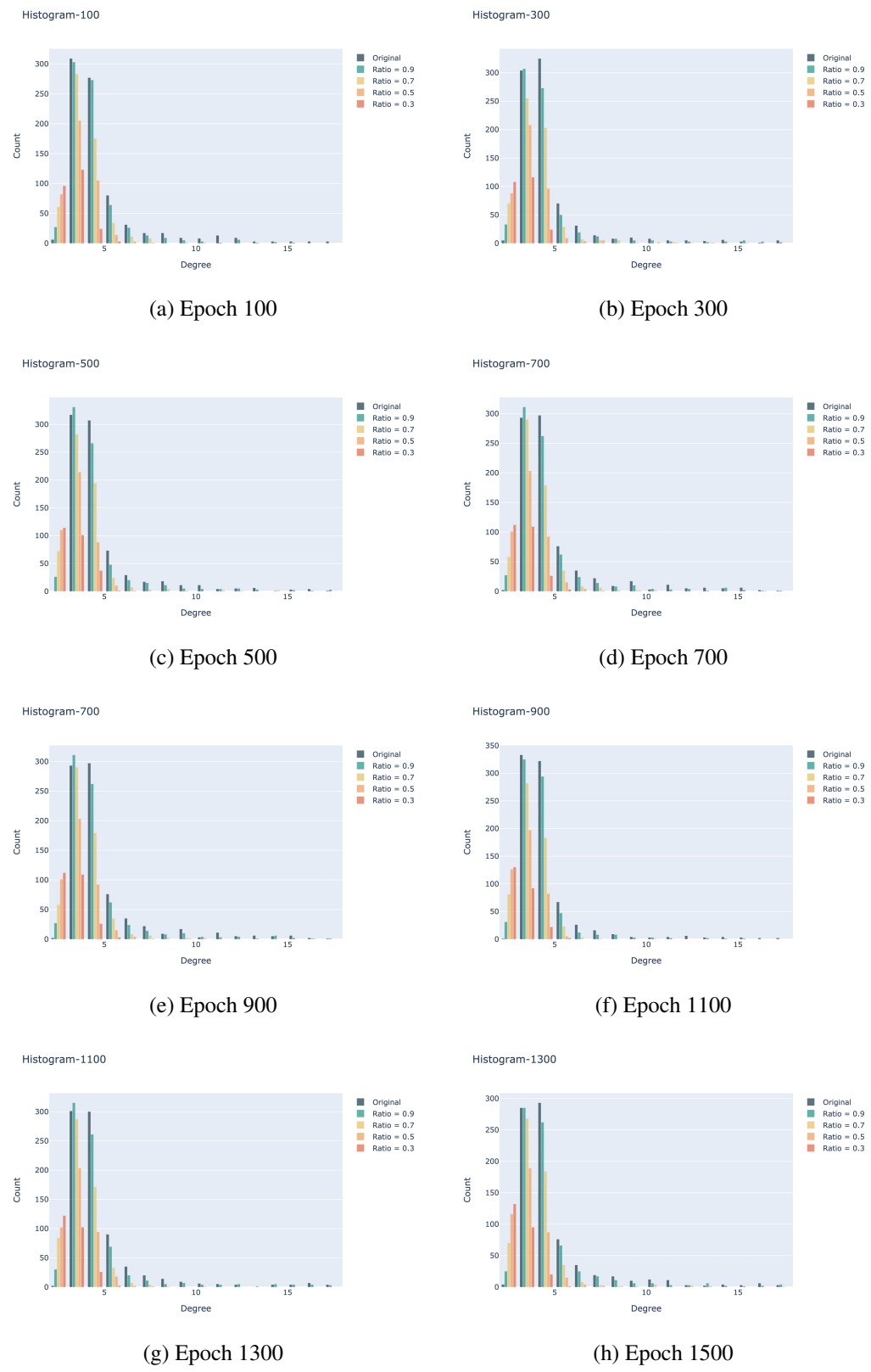

Figure 9: Degree Histogram of Flickr dataset at different epochs and pruning ratio

