# OpenReview forum: "LINGUINE: LearnIng to pruNe on subGraph convolUtIon NEtworks"
_ICLR.cc/2021/Conference — Reject_

### Official Review · AnonReviewer2 · 2020-10-27
**not convinced by the smart pruning algorithm**

**Rating:** 3
**Confidence:** 3

**Review:**

————————————————Summary————————————————
The author presented Linguine as a subgraph GCN training framework to train GCNs on large graphs. It includes bootstrapping and smart pruning algorithm to improve the quality of GCN models. In each forward pass, smart pruning algorithm prunes nodes to extract a concentrated smaller subgraph from the large subgraph randomly sampled previously.

————————————————Strengths————————————————
+ It is good to visualize the Pruning example.
+ It is good to include rich baseline methods.
+ well organization

————————————————Weaknesses————————————————
- I am not convinced regarding the nodes with high training loss as redundant nodes. From my understanding, redundant nodes mean the nodes carry redundant information. Apparently, nodes that are hard to learn by the existing models bring richer information rather than bring redundant information.
- GCN prone to overfit the error (error with low degree can easily get low training loss), the proposed methods will aggravate the negative impact brought by the error.
- From my point of view, the nodes with the highest degree are the most import node which affects message passing among nodes in the graph the most. However, it seems that smart pruning tends to prune these important nodes instead.

Questions:
Since vanilla GCN is non-convex, I don’t see why we need the proof of the theorem on the convergence on joint training scenario in the convex scenario.

---

> ### Author Response · Authors · 2020-11-17
> **Thank you very much for the help comments and review**
>
> First, we would thank the anonymous reviewer for their helpful comments. In this work, we used a purely empirical method to illustrate the nodes that were less informatic vs the nodes that were more important in bootstrapping. The work redundant might be confusing to the reader. We are saying that redundant nodes are less important rather than carrying redundant information.
> We would like to know the evidence showing this property of GCN and do a more extensive study on this phenomena, thank you for pointing this out.
>
> Although the message passing among nodes in the graph is the most important, they are not easy to learn from the empirical study as their pruning score is low. This means that their existence in the network is less significant than other nodes.

---

### Official Review · AnonReviewer4 · 2020-10-28
**This is an interesting work employing meta-learning setup for creating informative sub-graphs for training GCNs on large graphs. However, the paper requires major revisions and additional results to support its claim strongly.**

**Rating:** 3
**Confidence:** 4

**Review:**

This work proposes a new framework, called LINGUINE, to produce high-quality sub-graphs that can assist in effective training of graph convolutional networks (GCNs) with a lower computational cost. The framework uses two consecutive components - (1) Bootstrapping, which learns a meta-model that can assign weights to nodes by being jointly trained with a proxy ‘light’ GCN model (2) Smart pruning, which uses the learned meta-model to perform pruning on the large graph as the GCN model trains on it using the student-teacher learning concept. The paper presents results that the bootstrapping method improves the accuracy of smaller GCN models on large graph classification tasks. The smart pruning method removes nodes that are hard to learn or have a high degree of neighboring nodes in the subgraph, thus reducing error.

Strengths:
+ The idea of performing smarter subgraph selection via meta-learning is very interesting. Sub-graph selection is an important problem in the domain for training GCN models on large graphs
+ The paper presents a theorem to show that under a convex setting, the proposed join training during bootstrapping is guaranteed to converge
+ The benchmarking results illustrate that the bootstrapping step with a 2 layer SageNet architecture performs similar to the much bigger DeepGCN model on the OGBN-Product (Amazon product co-purchasing graph) dataset
+ On the OBGN-protein dataset, the results demonstrate that the smart pruning step can improve model performance by selecting sub-graphs without nodes  that are hard to learn or have a high degree of neighboring nodes
The paper also presents visualizations and results on batch size effect and pruning ratios on the model performance

Weaknesses:
+ My understanding from the method section is that LINGUINE is a framework that combines the meta-model learning and smart pruning to produce high-quality subgraphs that can be used to train a GCN effectively. However, the benchmarking results have been performed for the bootstrapping step only, instead of the whole framework. Since the output of this step (according to Algorithm 1) is meta-model, how are the F1-scores being reported in Table 1?
+ What is the architecture of the light-weight GCN? How is it used in reporting the bootstrapping results?
+ The paper states that the “sampling algorithm is user-defined”. Is this suggesting only a node sampling algorithm? Also, while it is mentioned that a random node algorithm is chosen, the results also have a random-walk algorithm. How do the two compare in terms of cost-effectiveness? Since one gives better performance of the other.
+ The reasoning for missing DeepGCN F1-scores for 4 datasets is not provided.
+ The paper describes results on the OBGN-protein dataset, for smart pruning. This is a different dataset from the ones for which the F1-score has been reported. The justification for this choice is missing. Similarly, the visualization results have been presented for Yelp data, and pruning ration graphs in the appendix for Flickr. These results should be presented consistently for one/multiple datasets.
+ For a paper trying to solve the computational cost issue of training on large graphs, there are no results comparing the numbers related to memory or time for LINGUINE with other baselines. The discussion on such a tradeoff would be very useful to the end-user.
+There are multiple typos and grammatical errors in the paper that would require thorough reading and revisions.

Minor comments:
- Figure 1 is not intuitive in demonstrating different sampling strategies
- In Figure 2, the role of the light-weight Graph is missing
- Is there a reason to choose accuracy instead of F1-score for Figure 3(a)

---

> ### Author Response · Authors · 2020-11-17
> **Thank you for the advice in revision of this paper**
>
> First, we would thank the anonymous reviewer's expertise and review of this work. Bootstrapping itself performs very well for the 2-layer GNNs. Our smart-pruning algorithm is designed for large models (e.g. DeepGCNs) to reduce the extra overhead of the Bootstrapping algorithm. We specify the light-weight GCN the same as 2-layer sparse vanilla GCN, which is the backbone of GraphSAINT but without the variance reduction procedure.
>
> DeepGCNs outperform the previous 2 layer varieties the most in the OGBN-protein dataset, which is also reported in their DeeperGCN paper. Thank you very much for the advice on our presentation, we will continue to improve the consistency of this visualization to make it more clear to the reader.
> The computational cost of this work is not fully explored, which will become an important part of future revision. Our writing will also be polished further.

---

### Official Review · AnonReviewer1 · 2020-10-29
**Good empirical results; design not well justified**

**Rating:** 4
**Confidence:** 5

**Review:**

--------------
Summary
--------------

This paper proposes an approach to compute GNNs on pruned subgraphs. The authors use a "meta-model" to learn a good node pruning strategy during training. Then the meta-model is used to generate pruned subgraphs during inference. The proposed pruning algorithm can be applied to various graph samplers. The authors evaluate such pruning on two simple samplers, random node and random walk samplers.  Some theoretical analysis is performed. However, the analysis is not quite convincing. Evaluation on several graph benchmarks show that the proposed LINGUINE framework achieves good accuracy using pruned subgraphs.

------
Pros
------

+ The paper is well-organized. Ideas are easy to follow.
+ Accuracy on standard benchmarks is promising.
+ Scalability of GNNs is an important topic. This paper performs meaningful exploration by trying to learn the pruning stategy.

-------
Cons
-------

- The theoretical analysis (Theorem 1) seems problematic and not that useful. First of all, Theorem 1 relies on strong assumptions on the objective function (convex; L-Lipschitz), and Theorem 1 basically just tells the convergence rate of the (almost) standard GD algorithm. Conclusion from Theorem 1 neither gives new insights into the proposed optimization procedure (not much more than standard GD), nor guides us to design better algorithms. Secondly, the proof of Theorem 1 seems problematic:
    * The first issue is that Theorem 1 talks about full-batch training, while the main algorithm of LINGUINE relies on random graph sampling. Theorem 1 should consider such stochasticity rather than completely ignoring it. I think due to the irregular graph structure, convergence analysis might be much harder than the current one presented with Theorem 1. Then, it would be valuable if we could get some insights into how the sampling algorithm and pruning could affect convergence.
    * The second issue is that, even with the full-batch setting, the detailed proof in the Appendix doesn't seem correct (note, the equation indices under this bullet point refers to the equalities / inequalities in the Appendix rather than those in the main text). 1). How would the assumption that $D$ and $M$ being independent guarantee $\langle\nabla_D, \nabla_M\rangle=0$? I think $\langle\nabla_D, \nabla_M\rangle$ can be any value since $D$ and $M$ contain free parameters. Also, if $M$ and $D$ does not have the same amount of parameters, the dot product is invalid since $\nabla_D$ and $\nabla_M$ don't even have the same dimension. 2). Inequality 4 (following Eq. 3 rather than Eq. 2, typo in the paper) should be "$\leq$" rather than "$\geq$". This is a significant issue since the last inequality of 4 and all derivations following Inequality 4 may not hold. 3). There are multiple typos in the proof that sounds confusing. E.g., in between Eq. 1 and Eq. 2, $\nabla f(x^t) - \nabla f(x^*)$ should be $\nabla_i f(x^t) - \nabla_i f(x^*)$, and the norm should be L-2 norm rather than L-$x$ norm. Below Inequality 4, $\Delta_t - \Delta_t$ should be $\Delta_t - \Delta_{t+1}$.
- Lacks justification on the proposed meta-model design / training.
    * Why do we need to alternatively optimize $M$ and $G_{light}$ (Algorithm 1)? Since we are only performing one-step gradient descent on $G_{light}$ or $M$ in each training iteration, and the full model ($M$ + $G_{light}$) is end-to-end differentiable, I don't see the benefit of alternating weight update between $G_{light}$ and $M$. It seems you can simply update $G_{light}$ and $M$ in a single back-prop pass. Also, the proof of Theorem 1 still holds by removing the part of alternating update (in fact, removing alternating update may improve convergence).
    * The connection to meta-learning is weak. It has been promoted as a main contribution that $M$ is derived from meta-learning. However, I find such connection a bit weak. I personally think $M$ as just an auxillary NN similar to that used by AS-GCN (Huang, 2018). There isn't really a "teacher-student" relationship. Arranging data from easy to hard is also missing in the design -- LINGUINE simply perform random sampling. Treating the pruning as a process of scheduling data from easy-to-hard seems to be a misuse of the meta-learning concept. In summary, I tend not to connect $M$ with meta-learning. $M$ is more like a sampling NN.
    * Meta-model $M$ ignores subgraph structure completely. It is not justified why $M$ can learn a good pruning strategy. It seems that the input to $M$ is just node-level information. How can you prune subgraphs effectively without even considering the topology / structure of the subgraph?
    * Can we really use $\mathcal{W}$ to achieve good pruning? It is stated that during inference, pruning score $\mathcal{W}$ is processed by top-k selection. I would imagine that a good model should try its best to reduce the information loss due to top-k selection. Then naturally, $M$ should output $\mathcal{W}$ with many close-to-zero entries. However, there is no design in the training algorithm to encourage such close-to-zero behavior of $\mathcal{W}$. If $\mathcal{W}$ can freely take any value, the role of $\mathcal{W}$ sounds just like node attention (similar to edge attention in GAT). Then I wonder if we can simply train a GAT and prune those edges with low attention values. How would LINGUINE compare to such pruning based GAT?
    * Lacks information during inference. Algorithms 1 and 2 seem to imply that $M$ takes the same input as $I(\mathcal{V})$. According to the first paragraph of Section 3.2.1, the information in $I(\mathcal{V})$ seems to be only available during training (e.g., loss of previous epochs). How can you retrieve such $I(\mathcal{V})$ during inference?
- Lacks justification on the claims of "scalability and lower bias". Regarding scalability: from the sample size shown in the experiments, it seems that LINGUINE requires its subgraph sizes to be as large as those of the baselines, if not larger (e.g., 12000 nodes). Even though after pruning, the subgraph sizes can be smaller, $M$ still needs to process all the 12000 nodes in each subgraph. More importantly, in a 2-layer GCN, such subgraph size may even be larger than the full 2-hop neighborhood size. Then how would it help to alleviate neighbor explosion? Regarding bias, I didn't seem to find a discussion on why LINGUINE reduces biases.
- Unclear experiment setup: there are some missing details about the GNN architecture (e.g., what are the hidden dimensions), and about the hyperparameter tuning procedure (i.e., the only hyperparameter available is the batch size listed in Table 3). Therefore, it may be hard to tell if the comparison is fair.

Minor issues (e.g., typos):
* Abstract: "unreal" should be "non-realistic"
* Meta-learning paragraph in Section 2: incomplete sentence "However, due to the strong assumption of the oracle's existence in machine learning. "
* Equations 3, 4 (in the main text) seem confusing. Should $w$ and $W$ come with subscripts? What's the difference between $w$ and $W$?
* Multiple typos in the proof of Theorem 1 (see above).

-----------------------------------
Recommendation: Reject
-----------------------------------
In summary, I think there are multiple major issues with the analysis and design choices, as detailed in the above "Cons" section.

--------------
Questions
--------------

Below is a summary of the questions raised in the "Cons" section.

1. Please clarify the proof of Theorem 1.
2. Please justify the design choices. i.e., alternativing optimization on $G_{light}$ and $M$; inputs to meta-model $M$; values of $\mathcal{W}$.
3. Please specify how to feed $I(\mathcal{V})$ during inference.
4. Please clarify the claims on scalability and biases.
5. Please clarify the connection with meta-learning.
6. Detailed experiment setup.

---

> ### Author Response · Authors · 2020-11-17
> **Thank you for the expert review**
>
> First, we would thank the anonymous reviewer for their expertise and helpful review, which would benefit the future revision of this paper. This paper still has many uncleared problems and is probably not so ready from the point that the reviewer has pointed out. The reason for not using direct optimization but an alternate way is to compare fairly against other algorithms, where the validation set is not used in training. The scalability lies in that we can prune the large subgraph into a smaller one since the computation cost of a deep GCN is very large. For the 2-layer light GCN, which we used as the meta-model, the cost is minor in comparison. This is the concentration property of this meta-model.
>
> Our work follows the concept of self-paced learning as well as learning to teach. Those methods are also included in the meta-learning literature since our meta-model is trained with meta-information instead of the feature of nodes (e.g. datasets). Recently, many new works about 'meta-learning' have emerged, such as MAML. MAML is still using the data in meta-training, which is the fundamental difference between the branch of self-paced learning, which was proposed much earlier.
>
> More details of our experiments will be added to the paper for better reproduction. Thank you for pointing this out.

---

### Official Review · AnonReviewer3 · 2020-10-30
**initial review**

**Rating:** 5
**Confidence:** 3

**Review:**

### Summary

The main idea of this paper is to improve the training of GCNs by smartly selecting subgraphs to train on in order to reduce memory consumption when running on large graph datasets. The proposed model consists of two stages:
1) A bootstrapping stage. In this stage, a lighter-weight GCN and a meta-model are alternately trained. They use a 2-layer vanilla GCN to speed up training. The meta-model is fed in statistics from the training process, and is used to assign pruning weights to the nodes in a graph. For each iteration, a training graph is first randomly subsampled, then the meta model is used to smartly prune it further.
2) A smart pruning stage. Now the meta-model is fixed, and a heavier-weight GCN is now trained on graphs that are pruned using the meta-model weights.

The model is evaluated on standard large datasets (Flickr, Reddit, Yelp, PPI), as well as the newer OGB-product and OGB-proteins datasets. Using random walk sampling as the base sampling algorithm, they achieve up to 1% absolute improvement over the baselines.

They also claim that redundant nodes will be removed by this process, and that nodes that have high training loss or that affect how their neighbors are classified (i.e. high-degree nodes) are more likely to be redundant. They then show an analysis on the OGB-proteins dataset to support this claim.

### Recommendation / Justification
The results aren't bad, but the ideas in this paper aren't terribly novel, as there are already so many other node / subgraph sampling algorithms out there to scale GNNs, and I don't feel like this one contributes anything new to this area. Also the paper is lacking in analysis of why the method works or why it is justified, and I find the OGB-proteins dataset analysis at the end to be unconvincing.

### Suggestions
For the pruning analysis, the pruned nodes plots, like Figure 3(b) and those in the appendix, are a bit hard to interpret. It might be better to show some histograms with the statistics of the pruned nodes vs those that end up in the final training graphs, as these would be easier to read.

---

> ### Author Response · Authors · 2020-11-17
> **Author response to the review**
>
> First, we would appreciate the expert review from the anonymous reviewer. The justification of this method is important and will become one of the important working direction for us in the future revision of this paper. Our contribution to the area is an empirical study on what nodes are more important than others in training GNNs. The answer to this problem is fundamental and intuitive in the future design of better sampling algorithms.
> We did the histogram on the last page of the appendix. Your concern on the dataset analysis of OGB-protein is helpful for us to present a more detailed empirical analysis of this method in revision in the future

---

### Decision · Program_Chairs · 2021-01-07
**Final Decision**

**Decision:**

Reject

**Comment:**

This paper presents an approach for training GCNs by learning to select subgraphs to train on to improve efficiency when transferring the model to larger graphs. In the proposed method, a meta-model and a light-weight GCN are trained iteratively in turns. Results are presented on medium-to-large graph datasets such as Reddit, Flickr, Yelp, PPI, and OGB-Product.

The reviewers agree that the method and the results are interesting, and that the topic addressed in the paper is important, but that the paper generally needs more work in terms of presentation, motivation, experimental evaluation, and theoretical analysis to meet the bar for acceptance, which the authors also acknowledge during their rebuttal.